

# Effects of chronic exposure to clothianidin on the earthworm *Lumbricus terrestris*

Kate Basley and Dave Goulson

School of Life Sciences, University of Sussex, Brighton, East Sussex, United Kingdom

## ABSTRACT

Although neonicotinoids are targeted at insects, their predominant use as a seed dressing and their long persistence in soils mean that non-target soil organisms such as earthworms are likely to be chronically exposed to them. Chronic exposure may pose risks that are not evaluated in most toxicity tests. We experimentally tested the effect of field-realistic concentrations of a commonly used neonicotinoid, clothianidin, on mortality, weight gain, and food consumption to assess the impacts of chronic exposure over four months on fitness of *L. terrestris* individuals. We undertook three separate experiments, each with different exposure routes: treated soil only (experiment A), treated food and soil combined (experiment B) and treated food only (experiment C). Mortality was negatively affected by exposure from treated soil only with greatest mortality observed in the groups exposed to the two highest concentrations (20 ppb and 100 ppb), but no clear effect on mortality was found in the other two experiments. When clothianidin was present in the food, an anti-feedant effect was present in months one and two which subsequently disappeared; if this occurs in the field, it could result in reduced rates of decomposition of treated crop foliage. We found no significant effects of any treatment on worm body mass. We cannot rule out stronger adverse effects if worms come into close proximity to treated seeds, or if other aspects of fitness were examined. Overall, our data suggest that field-realistic exposure to clothianidin has a significant but temporary effect on food consumption and can have weak but significant impacts on mortality of *L. terrestris*.

## INTRODUCTION

Neonicotinoids are the most widely used group of pesticides in the world (*Jeschke et al., 2011*). Their leaching potential and systemic properties mean that many non-target organisms in agricultural landscapes are likely to be exposed (*Goulson, 2013*), and their current prophylactic use on many arable crops presents a potential for large scale contamination of non-target areas. Neonicotinoids are often applied as seed dressings (*Jones, Harrington & Turnbull, 2014*), with typically 94% of the active ingredient applied to the crop seed entering the soil rather than the crop (*Goulson, 2013*). Residues of these compounds have been detected in soil more than three years after introduction via seed treatments (*Botías et al., 2016*). Clothianidin, a commonly used neonicotinoid, has a reported half-life of 148–1,155 days in aerobic soil, potentially exposing soil-dwelling

Corresponding author
Kate Basley, katebasley@gmail.com, K.Basley@sussex.ac.uk

organisms such as earthworms for extended periods of time (*Jones, Harrington & Turnbull, 2014*). It is this reported persistence that is amplifying the concern surrounding the impact of neonicotinoids on non-target organisms.

The application of agricultural products such as neonicotinoids for the protection of agricultural and horticultural crops has been shown to introduce these compounds to the drilosphere (the part of the soil which is influenced by earthworm secretions and castings), where the soil acts as a sink for agricultural products (*Givaudan et al., 2014*). Neonicotinoids can also compromise the function of soil organisms that contribute to soil fertility which may limit crop yield (*Moffat et al., 2016*). Their presence in the soil profile poses a hazard to resident worm populations, as the same neural pathways that are the target of neonicotinoids in pest species, are also present in earthworms (*Volkov et al., 2007*). Acting as Nicotinic Acetylcholine Receptor agonists, very low levels can significantly disrupt neural functioning in bees (*Piiroinen et al., 2016*), so if the negative effects of the neonicotinoid are similar to those for other non-target insects (*Pisa et al., 2014*), the worm's critically important role in the maintenance of soil properties could potentially be affected. Exposure can either be by direct physical contact with a treated seed or contaminated soil or soil water. Moreover, it is typical for earthworm species to ingest soil particles as they burrow, hence presenting an oral route of exposure to the compounds (*Pisa et al., 2014*).

The majority of studies investigating the impact of neonicotinoids on earthworms have focused on *Eisenia fetida*, with the range of reported lethal concentrations based on this species and little consideration given to the sensitivities of other species (*Pisa et al., 2014*). *E. fetida* are compost worms, and so are not typically found in areas where neonicotinoids are in use, preferring warm and moist habitats with a ready supply of fresh compost material. They are also claimed to be less sensitive to environmental toxicants than other earthworm species (*Dittbrenner et al., 2010*), and so results from these test species provide little insight into the potential impact of pesticides on earthworms in arable ecosystems. A recent review exploring the biochemical and molecular markers as indicators of the accumulation of pollutants, specifically pesticides, reported varying levels of biomolecules in different parts of the earthworm which indicated varying sensitivity of earthworms to different xenobiotics (*Tiwari et al., 2016*).

*L. terrestris* is commonly found in grasslands and lawns, especially when the ground is left undisturbed (*Sherlock, 2012*) and is more representative of species found on agricultural land and in field margin soils than *E. fetida* (*Nuutinen, Butt & Jauhiainen, 2011*); its widespread geographical range and frequently high abundance make it a special target for concern and study, since it is likely to play a major role in contributing to soil health (*Tomlin, 1992*).

Anecic worm species such as *L. terrestris* live and feed on the soil and are of particular benefit to arable soils where worms can constitute up to 80% of total soil animal biomass (*Pisa et al., 2014*). Their wide and deep-penetrating burrows open up the structure of compacted and clayey soils by enabling water infiltration (*Nuutinen, Butt & Jauhiainen, 2011*), and soil fertility is enhanced by the breakdown of plant litter and the mixing of this litter with the soil (*Pisa et al., 2014*). Physical disturbance of the soil by tillage and ploughing can have strong negative effects on the abundance of *L. terrestris* and so higher

**Table 1 Impact of neonicotinoid imidacloprid on *L. terrestris*.** Lowest effective concentration is the lowest concentration at which a significant effect was reported. 0, little or no measurable effect; –, moderate decrease; —, large decrease. Table adapted from *Pisa et al. (2014)*.

| Measured endpoint | Impact | Lowest effective concentration | Duration of exposure to contaminant | Study |
|---|---|---|---|---|
| Survival | 0 | 4 ppm | 14 days | *Dittbrenner et al. (2012)* |
| Avoidance | 0 | | | |
| Burrowing | – | 2 ppm | 7 days | *Dittbrenner et al. (2011)* |
| Feeding activity | – | 43 mg m$^{-2}$ | 6 weeks | *Tu et al. (2011)* |
| Abundance | – | | | |
| Body mass change | – | 0.66 ppm | 7 days | *Dittbrenner et al. (2010)* |
| Cast production | — | 0.66 ppm | 7 days | |
| Cast production | – | 1.89 ppm | 7 days | *Capowiez et al. (2010)* |
| Body mass change | – | 0.189 ppm | 7 days | |

population densities are often found in field margins which may act as source areas for the worm, supporting population growth within the field (*Nuutinen, Butt & Jauhiainen, 2011*). As these worms feed at the soil surface they are likely to be exposed to higher concentrations of pesticides as agrochemical concentration is often higher at the soil surface (*Chagnon et al., 2014*).

To date, the few studies that have considered the impact of neonicotinoids on *L. terrestris* have focussed only on the neonicotinoid imidacloprid (Table 1). Studies showed little or no measurable impact on survival at 4 ppm (*Dittbrenner et al., 2012*), but with a moderate decrease in body mass and large decrease in cast production being observed at 0.66 ppm (*Dittbrenner et al., 2010*). Cast production was negatively affected at 0.66 ppm and 0.189 ppm (*Capowiez et al., 2010*).

The authors are aware of no published studies that have investigated the impact of chronic exposure of clothianidin on *L. terrestris*. Clothianidin has recently become the most commonly used neonicotinoid in the UK (*Defra, 2016*), and is regularly used for seed, foliar, and soil treatments (*Jeschke et al., 2011*). One study of agrochemical toxicity to *E. fetida* found clothianidin to be the most toxic of 45 pesticides tested, with an LC50 value of 0.28 µg cm$^{-2}$ from a filter paper contact test. When tested in artificial soil for 14 days, clothianidin showed the highest intrinsic toxicity against *E. fetida* with an LC50 values of 6.06 (5.60–6.77) mg kg$^{-1}$ (*Wang et al., 2012*). A recent review investigating the impact of different types of neonicotinoids at varied concentrations on earthworm survival, reproduction and behaviour was conducted across different types of earthworm species; *Pisa et al. (2014)* concluded that individuals are at risk of mortality if they consume soil or organic particles of about 1 ppm for several days. It is possible that long-term low level concentration of neonicotinoids in soil may pose a higher risk to earthworms from sub-lethal effects than can be deduced from short-term toxicity tests, which typically last for a few days (*Pisa et al., 2014*). Here, we experimentally test the effect of field-realistic doses of clothianidin on mortality, weight gain, and food consumption to assess the overall impacts on fitness of chronic exposure of *L. terrestris* individuals.

## METHOD

### Soil contamination

The soil moisture content of both a sharp sand and a sterilised Kettering loam was taken using a TDR© 'FieldScout' soil moisture content probe. Kettering loam is used by many researchers as a reliable earthworm culture substrate and has been proposed as a standard medium for toxicology tests (*Lowe & Butt, 2005*). The loam was mixed with a sharp sand to make a more friable substrate in order to ensure that the soil was uniformly contaminated. Clothianidin stock solution (made up in water) was diluted as appropriate with more spring water (ASDA, own brand), then mixed with sand and finally loam to give a 70:30 loam: sand mix with a 25% moisture content (*Berry & Jordan, 2001*).

Treatment groups of 0 ppb (control), 1 ppb, 5 ppb, 10 ppb, and 20 ppb were based on clothianidin concentrations detected in soil collected from the field margins of conventionally farmed, neonicotinoid treated oilseed rape and winter wheat fields in the UK (Range: 2.25–13.3 ppb and 0.41–19.1 ng/g respectively; both 100% frequency of detections) (*Botías et al., 2015*). These samples were collected in the spring, approximately 10 months post-drilling of treated crops in fields undergoing conventional arable rotation. 100 ppb was used as a positive control. While these levels were used to replicate those present up to 40 weeks since seed drilling, it should be noted that levels of 270–440 ppb have been found in soil up to three days after a single clothianidin application (*Ramasubramanian, 2013*).

### Food contamination

Primary waste paper sludge from a paper recycling plant (Sittingbourne, Kent, UK) was mixed with brewer's yeast at a 25:1, carbon to nitrogen ratio following methods described in *Butt (1993)*, and used as a food source (referred to hereafter as "food"). To ensure homogeneous distribution of the clothianidin solution throughout the food, clothianidin stock solution was first added to spring water (ASDA own brand) and yeast before thoroughly mixing in the paper waste. Food was treated to the following levels: 0 ppb (control), 1 ppb, 5 ppb, 10 ppb, 20 ppb and 100 ppb.

### Microcosm set-up

Tops were removed from 180 × 4 litre plastic bottles (henceforth described as "microcosms") and they were each filled with 1.5 kg of contaminated soil substrate. Three separate experiments were set up: A—treated soil only, B—treated soil and treated food and C—treated food only, with 10 replicates per treatment group in each exposure and control group. Care was taken to ensure that no large air pockets were present as these could be used by the worms as a refuge. The bottle opening was covered with fine plastic mesh to prevent escape. Every microcosm received 70 g of food atop of a stainless-steel mesh (6 mm × 6 mm) placed on top of the soil substrate. The 720 worms were purchased from Worms Direct (Maldon, Essex, UK) and all were mature with clitellum. Prior to the experiment worms had been fed on leaves but all underwent a 7-day acclimatisation period where their food was swapped to the paper waste and yeast mixture used in this experiment. Each microcosm housed four worms. Experiment A received worms that were approximately two months older than individuals used to initiate experiments B and C. Microcosms were kept at 18 °C and 50–70% relative humidity, following *Lowe & Butt (2005)*.

## Data collection

Every four weeks, the contents of each microcosm was emptied into clean buckets, and the worms were gently washed and blotted dry. The worms from each microcosm were weighed together as a group; body mass has previously been shown to be a sensitive biomarker in the earthworm (*Dittbrenner et al., 2010*). In order to avoid additional stress to the individuals, worm weight was not standardised by voiding the gut contents of individuals prior to worms being weighed. The weight of food remaining on the grill was then subtracted from the starting weight each month and is henceforth described as 'food consumed'. However, it is important to note that some of this food had been taken down into each burrow and stored i.e., it had not actually been ingested by the worms. Cast production can be used as a proxy for earthworm activity (*Capowiez et al., 2010*), however, casts could not be separated from the food as worms had commonly cast directly into their food source. Obvious casts were removed from the edges of the grill before the remaining food was weighed. The worms were then placed back into the bottle with the same soil. The remaining food was discarded and replaced with freshly contaminated food and any water lost through evaporation from the soil (as defined by weight lost from a bottle of soil without worms) over the month was replaced in order to return the soil moisture to 25% (*Berry & Jordan, 2001*). Each experiment ran for four months in total.

## Data analysis

The average weight of individuals and the average amount of food consumed per worm were calculated every four weeks for each replicate. All analyses were carried out using SPSS version 22 (*IBM Corp, 2013*). Worm weights across treatment groups were compared using repeated measures ANOVA when assumptions of normality (as defined by the Shapiro–Wilk statistic) were met. The assumption of sphericity (as defined by Mauchly's statistic) was not met for data from any treatment group, therefore Greenhouse-Geisser adjustments were made to correct the ANOVA and it is this adjusted $p$ value that is reported. The within-subject variance of food consumed per worm was found to have significant heterogeneity and therefore non-parametric Kruskall–Wallis $H$ tests were preferred for this variable. Significant effects were investigated further using pair-wise comparisons using Dunn's procedure with a Bonferroni correction for multiple comparisons.

Survival curves were fitted to mortality data for each exposure group using the non-parametric Cox's proportional hazards model (CoxPH). The CoxPH assumes proportional hazards (chance of mortality) within treatment groups using control group mortality as a reference. The output from this model was compared with a parametric model, alternately assuming a constant hazard and a non-constant hazard with Weibull errors to ensure good model fit (*Rotheray, 2012*). To assess the effects of treatment on mortality, a separate CoxPH was fitted to compare pooled treatment groups with pooled control groups, applying any level of clothianidin treatment on effect on survival.

## RESULTS

### Experiment A: treated soil

Neither the weight or food consumed by worms kept in treated soil and fed untreated food varied significantly across treatment groups over time (weight: $F = 1.231$, $D.F = 11.1$,

$p = 0.279$, repeated measures ANOVA, food: week 4: $X^2(5) = 10.443$, $p = 0.064$; week 8: $X^2$ $(5) = 7.073$, $p = 0.215$; week 12: $X^2(5) = 3.817$, $p = 0.576$; week 16: $X^2(5) = 5.44$ $p = 0.364$, Kruskall–Wallis (Figs. 1A and 1B). Mortality was lowest in the control group across all time points, with 52% of the total population remaining at week 16 (Fig. 1C). The CoxPH detected a significant effect of treatment on mortality ($Z = 2.348$, $p = 0.0189$). However, there was no clear dose–response effect at higher doses (Fig. 2), with worms exposed to 20 ppb clothianidin having the highest mortality (80% by week 16).

## Experiment B: treated soil and treated food

There was no significant difference in the weight of worms between treatments when exposed to clothianidin in both soil and food (Fig. 3A) ($F = 1.825$, $D.F = 10.365$, $p = 0.062$). Analysis of food consumption revealed significant differences in consumption across treatment groups over time, with generally lower consumption when exposed to higher pesticide concentrations (week 4: $X^2(5) = 29.639$, $p \leq 0.001$; week 8: $X^2(5) = 34.876$, $p \leq 0.001$ and week 12: $X^2(5) = 11.650$, $p = 0.040$ but not week 16: $X^2(5) = 8.761$, $p = 0.119$). Pairwise comparisons highlight significant differences between all treatment groups and 100 ppb (Fig. 3B) at week 4 (adjusted p: 1ppb $\leq 0.001$, 5ppb $= 0.010$, 10ppb $\leq 0.001$, and 20 ppb $= 0.002$), and at week 8 between 100 ppb and 1, 5 and 20 ppb (adjusted $p \leq 0.001$, $<0.001$ and .005, respectively), there was no difference in consumption at weeks 12 and 16. Highest total mortality was observed in the 100 ppb treatment group (25% mortality) but the differences in mortality between treatment groups was not significant (CoxPH $Z = -0.173$, $p = 0.863$, Fig. 3C).

## Experiment C: treated food

There was no significant relationship between clothianidin concentration and weight for the worms fed on treated food only ($F = 1.809$, $D.F = 8.870$ $p = 0.078$). However, the amount of food consumed was significantly different across treatment groups at week 4 and week 8 ($X^2(5) = 35.304$, $p \leq 0.001$ and $X^2(5) = 11.241$, $p = 0.047$, respectively). Pairwise comparisons at week 4 show less food being consumed in the 100 ppb group than in other groups bar 20 ppb (adjusted p: 1 ppb $\leq 0.001$, 5 ppb $\leq 0.001$, 10 ppb $= 0.004$ and Control $= 0.007$), and less food consumed at 100 ppb compared to 1 ppb (adj. $p = 0.039$) at week 8. Mortality in Group C (Fig. 4C) was highest in worms fed with 20 ppb treated food (27.5% mortality) and lowest in food groups 1 ppb and 5 ppb (10% mortality). Overall, there was no significant difference in mortality between treatment groups (CoxPH $Z = 0.935$, $p = 1.522$). It is notable that mortality in experiments B and C was markedly lower than in experiment A which used older worms.

## DISCUSSION

Our findings suggest that at field-realistic doses, chronic exposure to clothianidin does not have a significant effect on worm weight but that contamination of food can significantly impact the amount of food consumed for up to two months after initial oral exposure, and may also increase worm mortality.

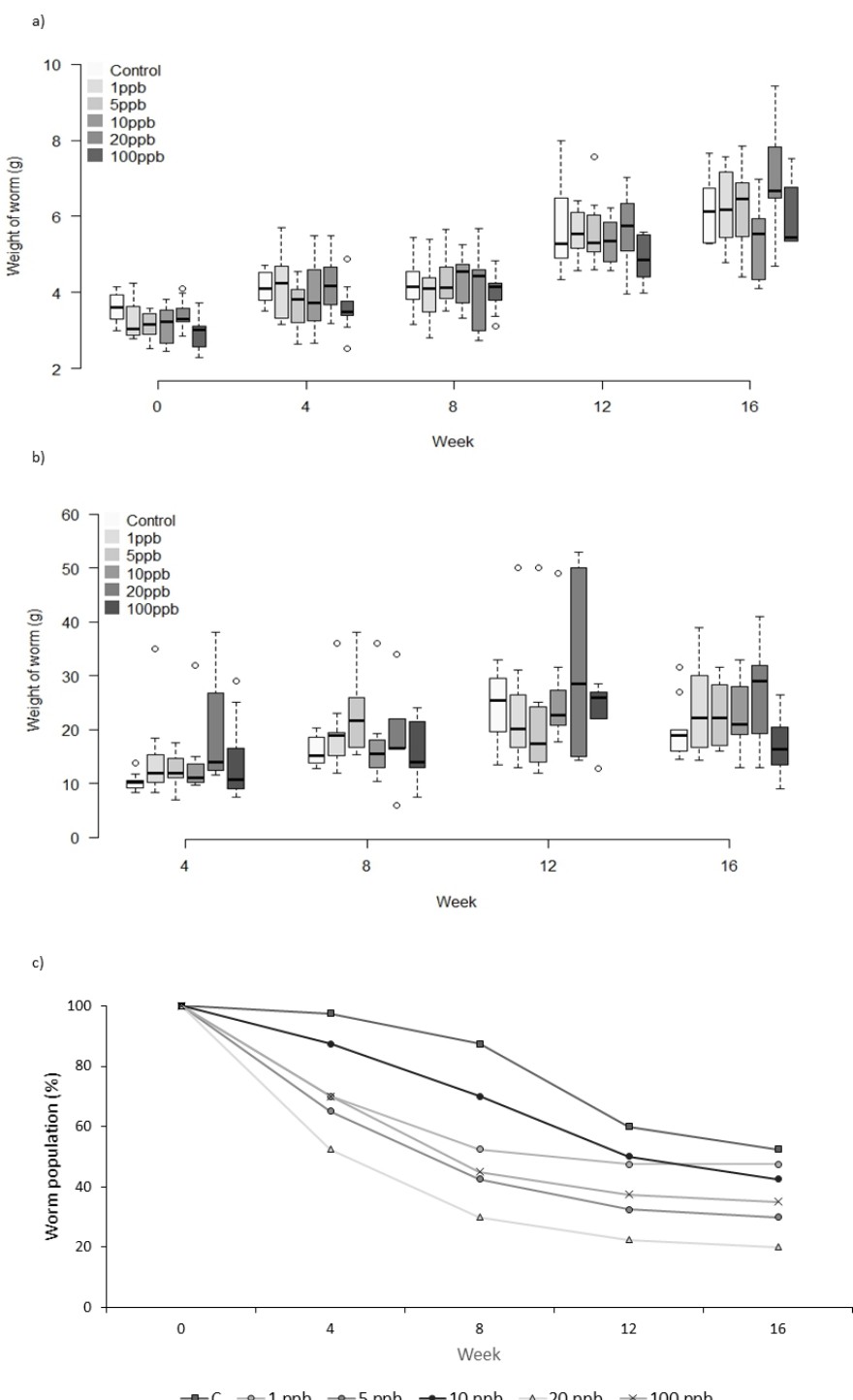

**Figure 1  Experiment A: clothianidin treated soil.** Changes in the mean weight (A) and mean food consumption (B) of *Lumbricus terrestris* individuals over time in clothianidin treated soil containing 1, 5, 10, 20, 100 ppb and control. Error bars show standard error of the mean. (C) Percentage of worms in relation to initial worm number.

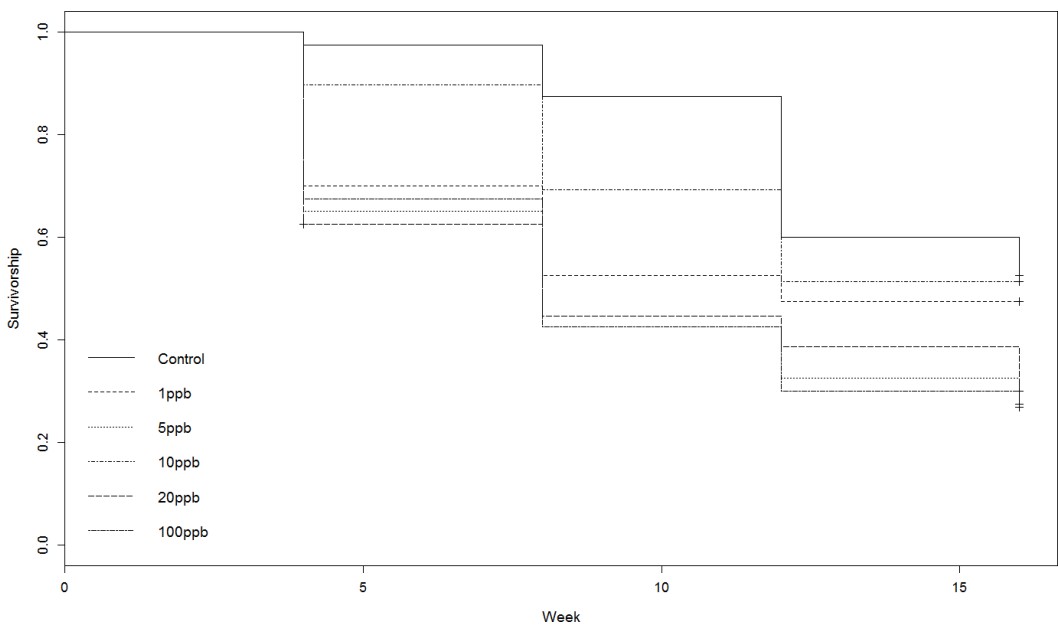

**Figure 2 Experiment A: clothianidin treated soil.** Cox proportional hazards model survival curve illustrating a significant effect of treatment on mortality ($Z = 2.348$, $p = 0.0189$).

## Mortality

Mortality levels for worms in experiments with treated food only, and treated food and soil suggest that clothianidin concentrations of $\leq 100$ ppb do not cause significant mortality above that of the control, whereas there was a significant effect of exposure to treated soil alone.

We speculate that this may be because the worms in experiment A were two months older than those in experiments B and C, but this would clearly require further investigation. Patterns of mortality across the three experiments were unclear as all lacked a clear dose–response effect. Of the 13 previous studies on the effects of neonicotinoids on earthworm survival that reported LD$_{50}$ values, only one studied clothianidin but used *E. fetida* as its study species. *Wang et al. (2012)* describe clothianidin as "super-toxic" to *E. fetida* (contact toxicity survival: 0.28 µg/cm, soil toxicity survival: LC$_{50}$ = 6.06 ppm) though this level is high compared to reported field concentrations and hence the phrase may be misleading. All other studies investigated imidacloprid or thiacloprid and reported LC$_{50}$ ranges from 1.5 to 25.5 ppm, with a mean of 5.8 and median of 3.7 ppm (*Pisa et al., 2014*). The longest exposure duration was six weeks, much shorter than the 16-week exposure used in this study. Further, seven of those 13 studies reported lowest effective concentrations ranging from 0.7 to 25 ppm, with a mean of 4.7 and median of 1 ppm (*Pisa et al., 2014*); all of which are concentrations that are an order of magnitude higher the levels used in this experiment.

Our study aimed to test effects of exposure to field-realistic concentrations. Overall, our data suggest that chronic exposure to concentrations of clothianidin up to 100 ppb in food and/or soil have, at worst, only weak effects on mortality of *L. terrestris*. However, it should be noted that our study involved homogeneous mixing of the clothianidin throughout the

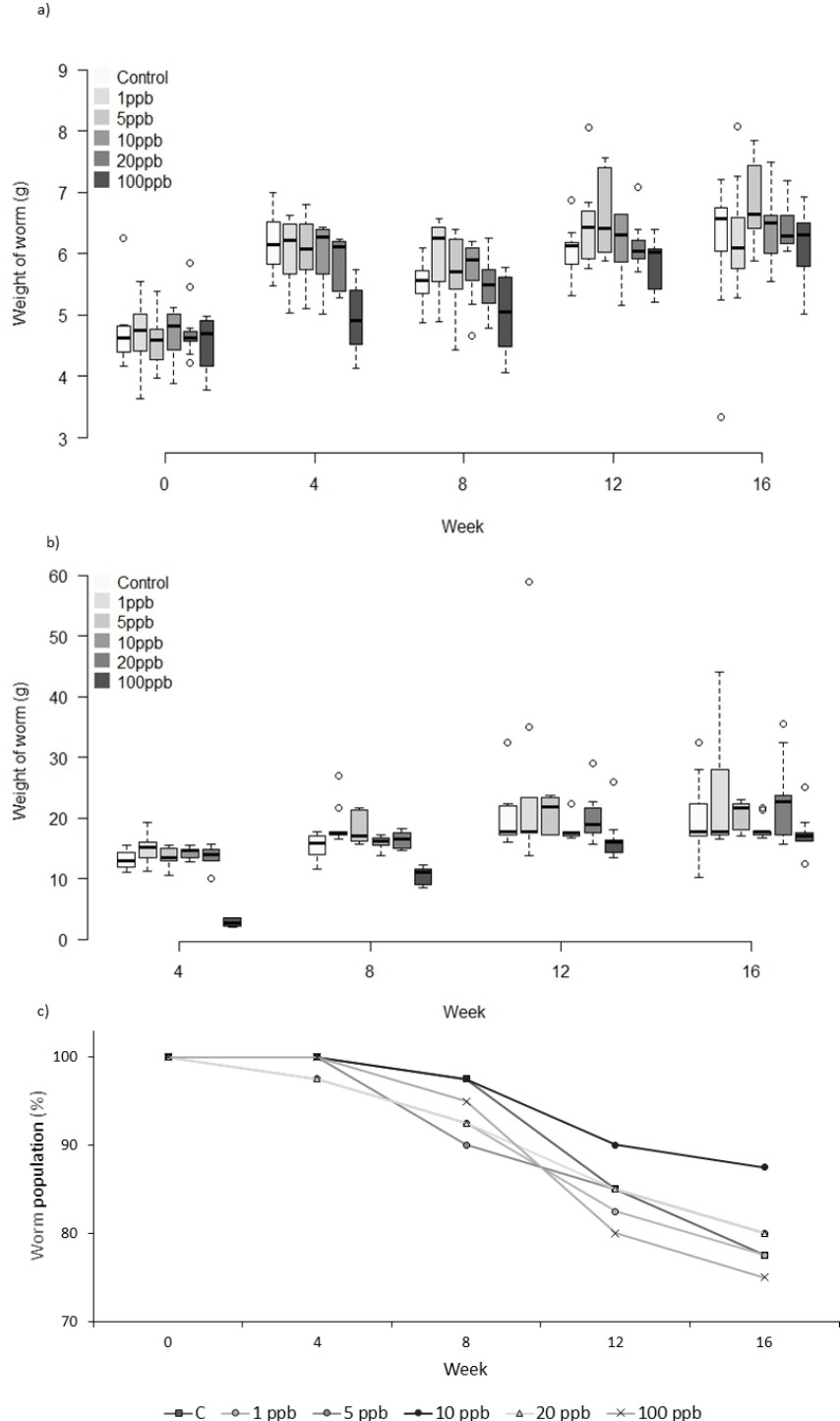

**Figure 3** **Experiment B: Clothianidin treated soil and treated food.** Changes in the mean weight (A) and mean food consumption (B) of *Lumbricus terrestris* individuals over time in clothianidin treated soil provided with clothianidin treated food containing 1, 5, 10, 20, 100 ppb and control. Error bars show standard error of the mean. (C) Percentage of worms remaining at each time point in relation to initial worm number.

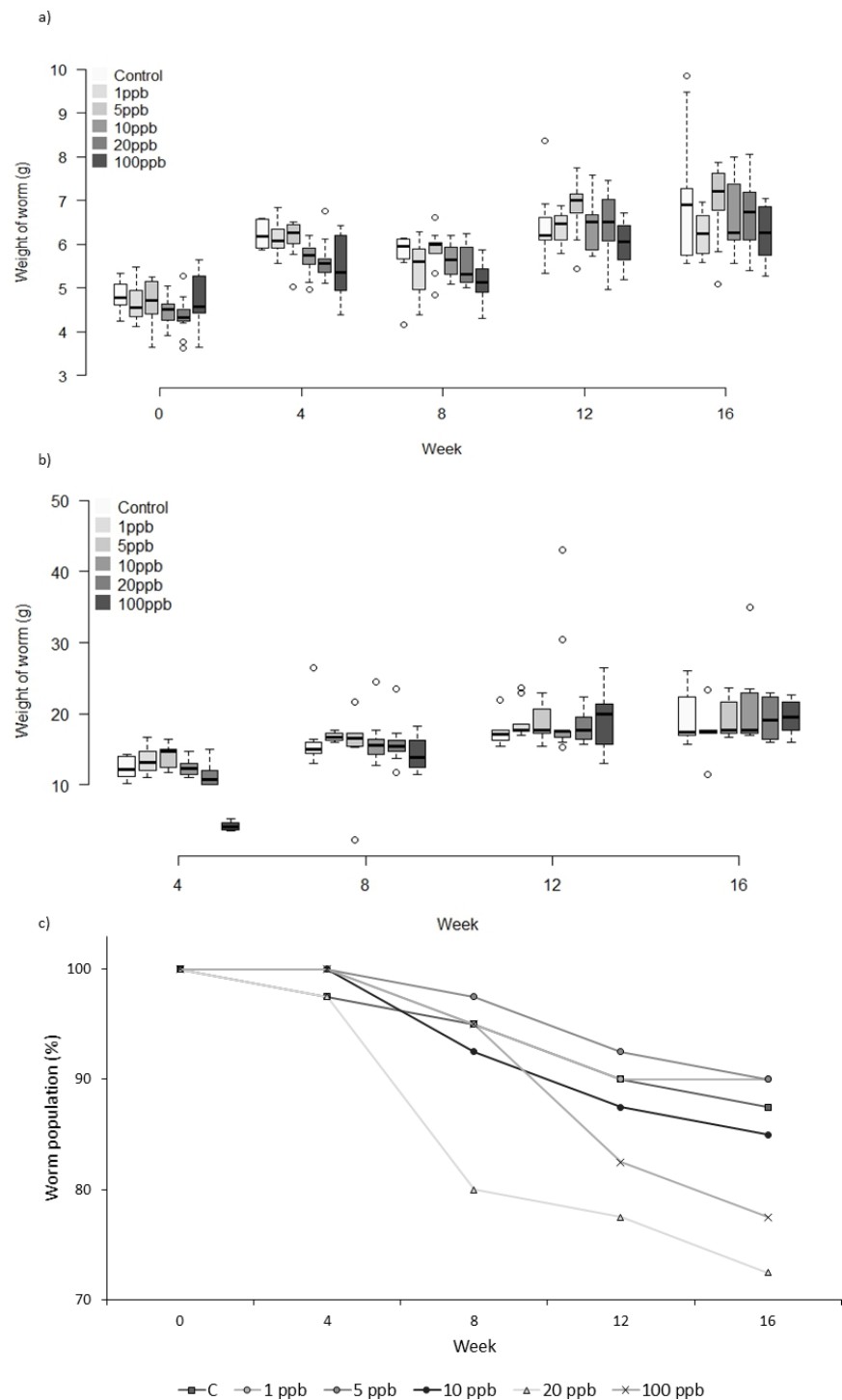

**Figure 4 Experiment C: Clothianidin treated food.** Group C: clothianidin treated food. Changes in the mean weight (A) and mean food consumption (B) of *Lumbricus terrestris* individuals over time in cloth-ianidin treated soil containing 1, 5, 10, 20, 100 ppb and control. Error bars show standard error of the mean. (C) Percentage of worms remaining at each time point in relation to initial worm number.

soil; it is possible that in a real-world situation worms may come across much higher levels of neonicotinoids by coming into close proximity to treated seeds or applied granules (*Pisa et al., 2014*).

## Weight

The presence of contaminants in soil may cause stress to the individual which can divert energy from reproduction, burrowing activity and growth (*Pelosi et al., 2014*). The use of body mass change as a biomarker is thought to be ecologically relevant, as high losses in body mass are thought to lead to negative effects on survival and reproduction (*Dittbrenner et al., 2010*). We found clothianidin to have no significant impact on body mass even over 16 weeks of exposure. Body mass in earthworms has not been used as a measured end point in experiments with clothianidin but comparison can be made with other neonicotinoids (*Pisa et al., 2014*). Three studies have monitored sub-lethal effects of imidacloprid on body mass and weight change in *L. terrestris* with lowest effective concentrations at 0.66 ppm (*Dittbrenner et al., 2010*), 0.189 ppm (*Capowiez et al., 2010*) and 2 ppm (*Dittbrenner et al., 2011*), all of which are higher than the treatments used in this experiment and above those generally found in the field. Our data suggest that field realistic exposure of *L. terrestris* to clothianidin does not impact on body mass.

As guts were not voided before weighing, it is possible that differences in worm weight between treatments could be masked or exaggerated by differences in gut content, for example if anti-feedant effects at high doses reduced gut content. However, we would expect this to lead to lower apparent mass at higher doses, which was not detected.

## Food consumption

In this study, we found clothianidin to have a significant negative effect on food consumption or food collection for the first two months of the experiment in groups where both the soil and food was contaminated and where only food was contaminated. We cannot discern whether the worms were able to detect and were repelled by the pesticide, or whether consumption reduced their subsequent appetite. A previous study with a different worm species, *Apporectodea spp.*, has shown that field-rate application of clothianidin (applied at 0.15 kg/ha) can retard long-term (four months) grass clipping decomposition (*Larson, Redmond & Potter, 2012*), a finding which our study corroborates. Reduced decomposition could potentially have long-term impacts on soil organic matter content which may be detrimental to crop growth.

An interesting feature of our data is the recovery of feeding rates towards the end of the experiment. As newly spiked food was provided every four weeks, this recovery is not because of a breakdown of clothianidin; it may be because the worms became desensitised, or because feeding inhibition was overridden by hunger. Food consumption recovered more quickly when only food was contaminated, compared to when both soil and food were contaminated, suggesting that both oral and contact exposure retards the recovery of the individual to a greater degree than oral exposure alone. Oligochaetes have been found to increase digging activity when exposed to thiamethoxam (*Alves et al., 2013*), so it is possible that any negative effect of exposure to clothianidin through treated soil is masked

due to an irritant effect: a worm's energy requirement may increase as a result of elevated activity caused by irritation, which may therefore increase food consumption.

Earthworms are known to be able to distinguish pollutants in soil, though it is not known if this behaviour is due to being able to discern the biological availability of pollutants or other factors (*Alves et al., 2013*). Our study design meant that individuals were unable to avoid the contaminated soil, and therefore laboratory exposure duration may not be representative of a typical field exposure duration; in a field-realistic scenario the individuals might be able to move away from contaminated soil, even though full field application of neonicotinoids is the norm. For example, they may be able to burrow deeper where contamination is likely to be lower. In this respect our experimental design may exaggerate effects compared to real-world situations. On the other hand, the results from this single chemical exposure experiment may not adequately reflect the full effect of the contaminant on *L. terrestris* as in field conditions they may often encounter multiple pesticides. Previous work on the impact of insecticidal chemistries on beneficial non-target arthropods and earthworms has shown there to be more significant effects from exposure to combination products than the singular components alone (*Larson, Redmond & Potter, 2012*).

The effects of neonicotinoid pesticides are often discussed assuming that all neonicotinoids act in the same way with regard to their target sites and their effects. However, individual neonicotinoids have been reported to have distinct binding to the nicotinic acetylcholine receptor (nAChRs) and therefore may pose differential risks to non-target organisms (*Moffat et al., 2016*). It would thus be unwise to assume that other neonicotinoids would have similar effects on earthworms to those that we describe for clothianidin (*Moffat et al., 2016*; *Dittbrenner et al., 2011*).

## CONCLUSION

Our results show that chronic exposure of *L. terrestris* individuals to clothianidin at concentrations up to 100 ppb has no significant long term effect on the weight of individuals but has significant negative impact on the amount of food consumed over a 2-month period. We also found some evidence of elevated mortality at higher doses in older worms. The eventual recovery of food consumption exhibited in individuals fed treated food highlights the importance of long-term chronic exposure studies; previous experiments have only tested the acute effects of neonicotinoid pesticides on *L. terrestris*, and have tended to use very high doses that may not commonly occur in the field. Although we cannot rule out negative effects on worms over longer periods, when in the immediate vicinity of treated seeds, or from combined exposure to neonicotinoids and other pesticides or stressors, our results suggest that exposure to soils and foodstuffs contaminated with field-realistic levels of clothianidin does not have lasting harmful effects on the growth or survival of younger *L. terrestris* individuals. Further work is required to investigate how worm age may influence their susceptibility to pesticides.

## ACKNOWLEDGEMENTS

We would like to thank John Lloyd for assistance with data collection and Dr Beth Nicholls for critical reading of the manuscript.

### Funding

This work was funded by the Natural Environment Research Council grant NE/K007106/1. The funders had no role in study design, data collection and analysis, decision to publish, or preparation of the manuscript.

### Grant Disclosures

The following grant information was disclosed by the authors:
Natural Environment Research Council grant: NE/K007106/1.

### Competing Interests

The authors declare there are no competing interests.

### Author Contributions

- Kate Basley conceived and designed the experiments, performed the experiments, analyzed the data, contributed reagents/materials/analysis tools, wrote the paper, prepared figures and/or tables, reviewed drafts of the paper.
- Dave Goulson analyzed the data, wrote the paper, reviewed drafts of the paper.

### Data Availability

   The raw data has been supplied as Supplemental Information 1.

### Supplemental Information

Supplemental information for this article can be found online at http://dx.doi.org/10.7717/peerj.3177#supplemental-information.

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
