# Peer review of "Effects of chronic exposure to clothianidin on the earthworm Lumbricus terrestris"

_PeerJ, doi:10.7717/peerj.3177_

## Round 0.1 · original submission · Minor Revisions

· Academic Editor

Minor Revisions

Your manuscript has been reviewed by three independent reviewers and, as you can see, they are recommending it publication. However, you have to consider the points raised by two of them. Please, be sure to send a tracked copy with the revised manuscript and a response letter to reviewers comments to speed the final acceptance of your manuscript.

·

Basic reporting

No comment.

Experimental design

Line 129: Describe the form of Clothianidin used. Commercial or technical grade formulation.

Validity of the findings

No comment.

Additional comments

I commend the authors for their large data set, moreover, the manuscript is clearly written in professional language.

·

Basic reporting

Well written and structured manuscript.
The introduction is clear and up to date.
Clear presentation of the results (even if no all of them are easy to interpret).

Points requiring improvements:
- better explanations about "realistic" CLO concentrations expected in the field. Authors provide a reference (L134) but it is better if the reader has direct access of the lmain findings of this cited reference (a range of concentration observed or frequency of detection).
- Better explain why sand was mixed with the loamy soil (not sure sandy soils are current for crops) L130
- could you ensure paper sludge is free of contaminants (or repellent ingredients) ? Presence of chlorine? Ink? L142

Experimental design

Innovative design (soil or food or both contaminated) enabling to discuss exposure route in earthworms.
Long-term exposure leading to (more) relevant results.

Some points need to be improved:
- owing the initial variability for eathworm weight (see Figure 1 - date 0), why not computing weight losses in percentage (of the initial biomass)? It is important to show raw values in the Figure but statistical analyses could be applied on percentages. If seeds are treated, it is likely that soil contamination would be highly variable, isn' it ?
- it is not clear whether food consumption, at each date, is computed on the initial earthworm abundance or only on remaining ones (better in the latter case and, if not, could explain some variability).
- Did the grill prevent earthworm to take food and to buty it in the soil (as currently obseved for leaves or straw) ? If not, then one should add that food was eaten (or buried), see for exampel L293. However this did not modify the main conclusions.

Validity of the findings

The (balanced) conclusions are supported by the data.
I particularly liked the way the authors did not try to over-interpret their findings (as often observed in ecotoxicology).

some details :

L250 : it sounds strange. Why did the authors expect higher effects when both soil and food were contaminated. Do not forget that earthworms (excepted epigeic) eat the soil. Moreover it is often observed that thay ate more soil when food is scarce ….

L275 : since guts were not voided, pay attention that biomass variations can also be due to eathworm avoiding to feed (limited activity).

Figure 1 : I would comment the very high variability observed for food consumption at 40 ppb... this trend is often observed just below the sublethal concentration.

Additional comments

Very nice manuscript. This will add to the litterature.

·

Basic reporting

.

Experimental design

Appropriate and satisfactory

Validity of the findings

Appropriate and satisfactory

Additional comments

Reviewer’s Comments
The authors Basley and Goulson in their manuscript, ‘Effects of chronic exposure to clothianidin on the earthworm
Lumbricus terrestris’ have presented some evidence-based recommendations. They have experimentally tested the effect of field-realistic concentrations of neonicotinoid, clothianidin, on mortality, weight gain, and food consumption to assess the impacts of chronic exposure over 4 months on fitness of L. terrestris individuals. They have conducted three separate experiments, each with different
exposure routes: treated soil only (experiment A), treated food and soil combined (experiment B) and treated food only (experiment C). They observed negative impact by exposure from treated soil only with greatest mortality. They have observed stronger adverse effects if worms come into close proximity to treated seeds. They have concluded that field-realistic exposure to clothianidin had a significant but temporary effect on food consumption and could exert weak but significant impacts on mortality of L. terrestris.

The manuscript may be approved for publication subject to revision suggested as following;
1. There are some typos that must be eradicated. There are words fused with each other and proper spacing is also missing. The ethical issues must be addressed. Only relevant references are required to be cited. The authors are required to strictly adhere with the format of the journal while drafting their manuscript starting from the affiliations and afterwards.
2. The Introduction, Discussion and Reference sections may include the recently published review article on the subject shown as following;
Tiwari, R. , Singh, S. , Pandey, R. and Sharma, B. (2016) Enzymes of Earthworm as Indicators of Pesticide Pollution in Soil. Advances in Enzyme Research, 4, 113-124. doi: 10.4236/aer.2016.44011.

3. An account of the temperature, humidity, and soil composition may be added to make the observations and discussions more realistic.

4. The authors should have added the underlying biochemical mechanisms about actions of the xenobiotic used in the present research responsible for negative or no impact on mortality of earthworms.

---

## Round 0.2 · accepted · Accept

· Academic Editor

Accept

Dear Kate, thank you for your revised version.

I have looked at the text and in your letter and, regarding the question/statement in the rebuttal letter: "This is an interesting observation, but we are unable to find anywhere in the literature where such an effect is described. If the referee were able to supply a reference we would happily include it." Do you want to add any additional reference? If so, let me know and I will ask the journal to contact the reviewer.

And regarding the question two of Yvan, ...The sterilised soil had a very high clay content and so in order to make it more friable we added some sand. We felt this was necessary in order to ensure that the soil was uniformly contaminated (the clay could not easily be mixed).... The second sentence is not in the manuscript. I realize that it could be introduced in text, because this information will be important to others that will use similar approaches of exposure to chemical agents.